# A New Study on the Structure, and Phase Transition Temperature of Bulk Silicate Materials by Simulation Method of Molecular Dynamics

Dung Nguyen Trong [1,2,*], Van Cao Long [1], Ştefan Ţălu [3,*], Umut Saraç [4], Phu Nguyen Dang [5] and Kien Pham Huu [6]

1   Institute of Physics, University of Zielona Góra, Prof. Szafrana 4a, 65-516 Zielona Góra, Poland
2   Faculty of Physics, Hanoi National University of Education, 136 Xuan Thuy, Cau Giay, Hanoi 100000, Vietnam
3   The Directorate of Research, Development and Innovation Management (DMCDI), Technical University of Cluj-Napoca, 15 Constantin Daicoviciu St., 400020 Cluj-Napoca, Romania
4   Department of Science Education, Bartın University, 74100 Bartın, Turkey
5   Faculty of Electronics and Telecommunications, VNU-University of Engineering and Technology, Hanoi 100000, Vietnam
6   Thai Nguyen University of Education, no. 20 Luong Ngoc Quyen, Thai Nguyen 24000, Vietnam
*   Correspondence: dungntdt2018@gmail.com (D.N.T.); stefan.talu@auto.utcluj.ro (Ş.Ţ.)

**Abstract:** In this paper, the structure and phase transition temperature of bulk silicate materials are studied by the simulation method (SM) of molecular dynamics (MD). In this research, all samples are prepared on the same nanoscale material model with the atomic number of 3000 atoms, for which the SM of MD is performed with Beest-Kramer-van Santen and van Santen pair interaction potentials under cyclic boundary conditions. The obtained results show that both the model size (l) and the total energy of the system ($E_{tot}$) increase slowly in the low temperature (T) region (negative T values) at pressure (P), P = 0 GPa. However, the increase of l determines the $E_{tot}$ value with very large values in the high T region. It is found that l decreases greatly in the high T region with increasing P, and vice versa. In addition, when P increases, the decrease in the $E_{tot}$ value is small in the low T region, but large in the high T region. As a consequence, a change appears in the lengths of the Si-Si, Si-O, and O-O bonds, which are very large in the high T and high P regions, but insignificant in the low T and low P regions. Furthermore, the structural unit number of $SiO_7$ appears at T > 2974 K in the high P region. The obtained results will serve as the basis for future experimental studies to exploit the stored energy used in semiconductor devices.

**Keywords:** bulk $SiO_2$; low-temperature; high-temperature; pressure; phase transition temperature; molecular dynamics; structure

## 1. Introduction

Today, with the significant development of computer science and materials science for technology, it is possible for researchers to approach materials at the nanoscale. Among the study tools used, the simulation method (SM) of molecular dynamics (MD) is currently the most effective method for studying the structure, phase transition, and determining the phase transition temperature (Tm) of new materials [1]. In the framework of this method, the motion of atoms described by Newton's law of equations is studied.

In recent years, different scientists have successfully explored the influencing factors such as temperature (T) and pressure (P) on the structure and phase transition process in oxide materials (such as $CaSiO_3$ [2], $MgSiO_3$ [3,4], and bulk $Fe_2O_3$ [5,6]). The obtained results show that when T is rapidly reduced, the material moves to an amorphous state. Conversely, the material moves to a liquid state when T is increased. So a question arises: what will happen with $SiO_2$ material, and the characteristic quantities of its structure, or with $T_m$: do these ones follow the same rules as the above materials or not? To answer

this question, in this research we have focused on the structural characteristics and $T_m$ of bulk Silica (bulk $SiO_2$) materials by taking a proper pair interaction potential between the related components. It has been shown that the choice of van Beest-Kramer-van Santen (BKS) potentials introduced in Ref. [7] is the most proper one, because in different modified versions it reproduces the structural properties of bulk $SiO_2$ well [8]. Furthermore, it describes the dynamical properties of considered materials which are consistent with experimental data obtained before [8,9].

Bulk $SiO_2$ materials exist mainly in the amorphous (Amor) state as a powder or colloidal and are used widely in life. It is considered as one of the most important materials in the electronic industry. The effects of T and P on the heterogeneous kinetics of bulk $SiO_2$ material have recently attracted a lot of researchers' attention, in particular, the influence of T has become one of the focal points of their research [10–13]. By experimental methods (EMs) and also by SMs, sometimes by the Ab-initio method, authors of many previous publications have determined, among others, the length of links and the link angles in $SiO_2$ compounds [14–41]. Some papers have combined the EM with the SM of MD to clarify the change in the structural unit number of $SiO_2$ when P increases [42–44].

Generally speaking, the numerous results obtained before by different methods are very rich, but rather frequently are not consistent one with another. Here a question arises with regard to how the effects of T and P on the structure and phase transition of $SiO_2$ can be considered systematically. In this research, the answer to this question will be given by considering these effects in the framework of the SM of MD, and for that, as it has been emphasized above, the choice of BKS potentials is the most suitable. Therefore, in this study, the structural characteristics and phase transition of $SiO_2$ in the high T and P regions have been investigated. The results presented below provide a conclusion that the structural units $SiO_4$, $SiO_5$, and $SiO_6$ in the center of the earth do not exist for $SiO_2$ material. These are new results which could serve as a basis for future experimental studies.

## 2. Computational Methods

Initially, the sow randomly 3000 atoms of bulk $SiO_2$ (1000 atoms Si and 2000 atoms O) into the cube bulk model with the size (1) as expressed in Equation (1):

$$\rho = \frac{N}{V} \rightarrow l = \sqrt[3]{\frac{N}{\rho}} = \sqrt[3]{\frac{m_{Si}n_{Si}+m_{O}n_{O}}{\rho}} \qquad (1)$$

In this formula $m_{Si}$ = 26.98154, $m_O$ = 15.999. The model with the force field was expressed by the BKS pair interaction potential (according to the Equation (2)) and a periodic boundary condition in the framework of the SM of MD is proposed [15–19,23,45]:

$$U_{rj}(r) = \frac{q_i q_j}{r_{ij}} + A_{ij}e^{-B_{ij}r_{ij}} - B_{ij}r_{ij} - C_{ij}r_{ij}^{-6} \qquad (2)$$

whereas $n_{Si}$, $n_O$, $\rho$, $r_{ij}$, $q_i$, $q_j$, are molecular weights, the atomic numbers of Si, O and atomic density, distance links, and charges of the atoms i and j, correspondingly. $A_{ij}$, $B_{ij}$ and $C_{ij}$ are the potential coefficients of the model given in Table 1.

**Table 1.** Parameters of the bulk $SiO_2$ material [46,47].

| $SiO_2$ | Si-Si | Si-O | O-O |
|---|---|---|---|
| $A_{ij}(eV)$ | 0 | 18,003.5773 | 1388.773 |
| $B_{ij}(Å^{-1})$ | 0 | 4.87318 | 2.76 |
| $C_{ij}(eVÅ^5)$ | 0 | 133.5381 | 175.0 |
| $q_{i,j}(e)$ | - | $q_{Si}$ = +2.4 | $q_O$ = −1.2 |

By Verlet algorithm [48], one can determine the coordinates, velocity, and energy of atoms in the simulation process (Table 1). The authors create $SiO_2$ materials by running the $2 \times 10^4$ steps recovery statistics NVT (constant atomic number, volume, temperature), and $2 \times 10^4$ steps NVP (constant atomic number, volume, pressure) at T = 7000 K. The obtained result shows that atoms do not stick together and when the T is lowered from T = 7000 K to T = 300 K at P = 0 GPa. It can be noticed that the system is stable and reaches equilibrium at T = 300 K, P = 0 GPa. In the next step, both T and P are changed in the following way: first, the T of samples is increased from T = 300 K to T = 500 K, 1500 K, 2500 K, 3000 K, 3500 K, 4500 K, 5500 K, 7000 K at P = 0 GPa; in the next step P is increased from P = 0 GPa to P = 5, 10, 15, 20 GPa at T = 70 K, 300 K, 1273 K, 2974 K, 3500 K. After sample stabilization at the desired T and P, all samples run simultaneously with $5 \times 10^5$ steps NVE (constant atomic number, volume, energy) to the moment when the samples achieve equilibrium. To study the heterogeneous kinetics of bulk $SiO_2$, the samples are analyzed by radial distribution function (RDF) (according to Equation (3)) [48–54]:

$$g(r) = \frac{V}{N^2} \left\langle \frac{\sum_i n_i(r)}{4\pi r^2 \Delta r} \right\rangle \tag{3}$$

with g(r), V, N, $n_i(r)$, r is RDF, volume, atoms number, and coordinates denoting respectively the links (Equation (4)) [45]:

$$CN = 4\pi\rho \int_0^{r_1} g(r) r^2 dr \tag{4}$$

where CN, $r_1$ is the average Coordination Number (CN), the first peak position of RDF, and the bond angle. The relationship between the O-Si-O bond angle is used for the link lengths (applied for the links O-O, Si-O, Si-Si), and is computed by the following expression (Equation (5)) [45]:

$$\cos \alpha = \frac{2r_{Si-O}^2 - r_{Si-Si}^2}{2r_{Si-O}^2} \tag{5}$$

where: $\alpha$ = O-Si-O for the model defined at T and P. Also, during the heating process of $SiO_2$, it is evaluated according to the Nosé-Hoover formula [55,56].

To confirm the accuracy of the results, our results were compared with those obtained previously under the same T and P conditions. All these results during heating, P change, and model annealing were determined using the Nosé-Hoover formula [55,56], and the Verlet algorithm, and were run on the computer central system of the Institute of Physics, the Department of Physics & Astronomy of Zielona Gora University, Poland.

## 3. Results and Discussion

### 3.1. The Structural Characteristic Quantities

To study the structural properties of the bulk $SiO_2$, we study the bulk $SiO_2$ at T = 300 K and P = 0 GPa, and the obtained results are shown in Figure 1.

The obtained results show that the bulk $SiO_2$ at T = 300 K has a cube shape and is composed of two types of atoms: Si, and O (Si atoms are dark blue, and O atoms are red). It can be observed that Si atoms are not homogeneously distributed at the shell, while the O atoms are homogeneously distributed in the core layer (Figure 1a); the structural units number of $SiO_2$ is $SiO_4$ (green-blue), $SiO_5$ (red), $SiO_6$ (black) (Figure 1b). The parameters of atomic links for the links Si-Si, Si-O, O-O are calculated from the RDF. The links lengths are $r_{Si-Si}$ = 3.16 Å, $r_{Si-O}$ = 1.62 Å, $r_{O-O}$ = 2.64 Å which correspond to the first peak positions of RDF $g_{Si-Si}$ = 4.47, $g_{Si-O}$ = 24.76, $g_{O-O}$ = 4.78 (Figure 1c), whereas the average CNs are $CN_{Si-Si}$ = 4.11, $CN_{Si-O}$ = 4.01, $CN_{O-O}$ = 8.51 (Figure 1d). In addition, the quantities that characterize the structure of bulk $SiO_2$ also have the structural unit number, and the bond angle between atoms is 2975 atoms $SiO_4$, 121 atoms $SiO_5$, 7 atoms $SiO_6$, and the angles of the links is O-Si-O is 105°. The effect of low T (T < 273 K) corresponds to liquefied gases such as T = 4.22 K (helium), 70 K (nitrogen), 83.8058 K (argon), 90 (oxygen), 194.5 K

(carbon), high T (T > 273 K) and P at the respective T values will be studied in detail in the following section.

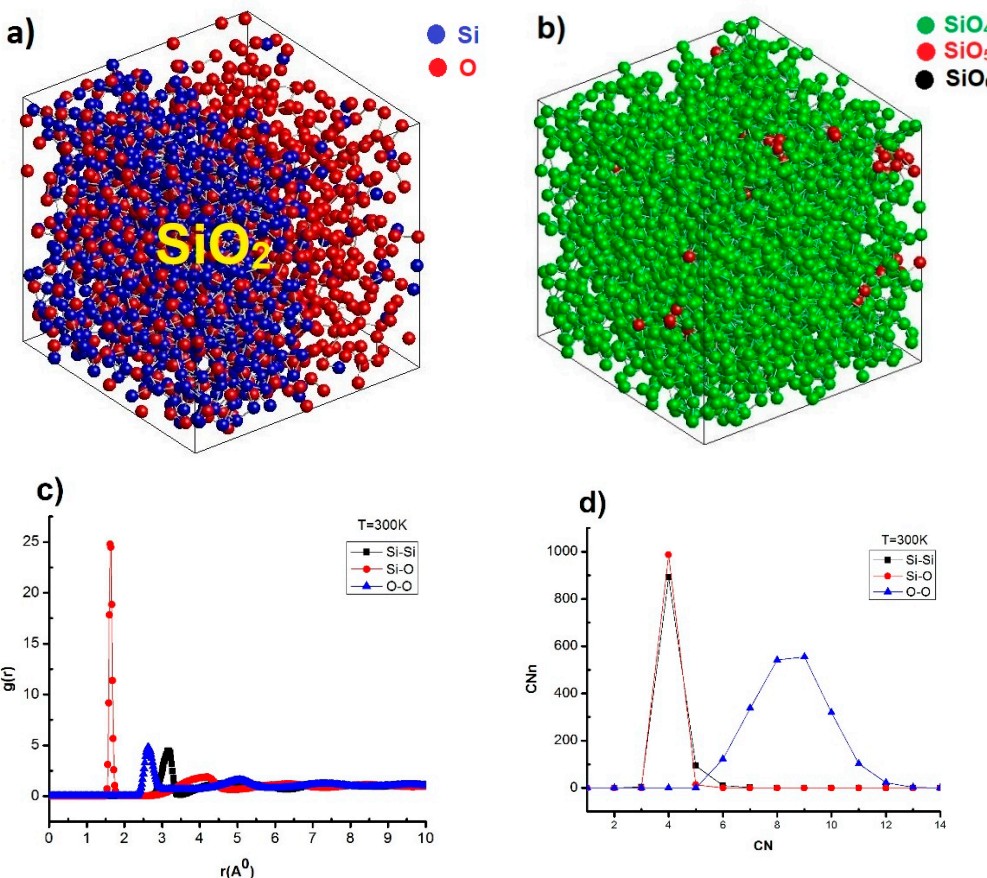

**Figure 1.** Structural characteristics of bulk SiO2 at T = 300 K, P = 0 GPa: Shape (**a**), the structural unit num-ber (**b**), the RDF (**c**), the CN (**d**).

*3.2. Effect of T*

3.2.1. High T Region

The results structural characteristic quantities of the bulk $SiO_2$ are presented in Figure 1, Table 2.

The obtained results show that the bulk $SiO_2$ has structural characteristic quantities at T = 300 K (Figure 1). When the T increases from T = 300 K to T = 500 K, 1500 K, 2500 K, 3000 K, 3500 K, 4500 K, 5500 K, and 7000 K, then the length (r) of links Si-Si, Si-O, O-O is $r_{Si-Si}$, $r_{Si-O}$, $r_{O-O}$ change as $r_{Si-Si}$ changes (in the range from $r_{Si-Si}$ = 3.12 Å to $r_{Si-Si}$ = 3.16 Å), $r_{Si-O}$ changes (from $r_{Si-O}$ = 1.56 Å to $r_{Si-O}$ = 1.64 Å), and $r_{O-O}$ changes (from 2.64 Å to 2.66 Å at T = 3000 K). Also, the length of the Si-Si, and Si-O bonds decreases greatly at T > 5500 K with $r_{Si-O}$ = 1.56 Å, which proves that $SiO_2$ has completely liquefied and has long fracture links which are caused by the size effect caused. The obtained results for $r_{Si-Si}$ are consistent with the results given by other authors using the SM [12–19,23] and the EM [21,22]. Similarly, one can also see accordance with other results for other links Si-O in link length obtained by SM [15–19,23] and by EM [12–14,20,21], whereas for the links O-O by SM [12–19,23] and [20,21] by EM (Table 2). It follows that the effect of T on the link is the length and the number of coordinates are negligible, which is a consequence of the form of g(r). This leads to the question of whether there is any other cause that strongly affects the structure of bulk $SiO_2$. The obtained results show that in the interval from T = 2500 K to T = 3000 K, $g_{Si-O}$ decreases slowly, and the $r_{O-O}$ increases suddenly, so it can be concluded that in this T zone, the phase transition from an amorphous state to a liquid state for bulk

$SiO_2$ has been realized. To study this phase transition, the number of structural units at different T values has been considered. The results are presented in Table 3.

**Table 2.** The structural characteristic quantity of bulk $SiO_2$ at different T values.

| T(K) | Links Lengths $r_{ij\,(\text{Å})}$ | | | First Peak Positions $g(r_{ij})$ | | | CN | | |
|---|---|---|---|---|---|---|---|---|---|
| | Si-Si | Si-O | O-O | Si-Si | Si-O | O-O | Si-Si | Si-O | O-O |
| 300 | 3.16 | 1.62 | 2.64 | 4.47 | 24.76 | 4.78 | 4.11 | 4.01 | 8.51 |
| 500 | 3.16 | 1.62 | 2.64 | 4.37 | 20.69 | 4.41 | 4.16 | 4.02 | 7.52 |
| 1500 | 3.14 | 1.64 | 2.64 | 3.62 | 12.72 | 3.54 | 4.23 | 4.02 | 4.02 |
| 2500 | 3.16 | 1.62 | 2.66 | 3.05 | 9.72 | 2.92 | 4.23 | 4.03 | 8.14 |
| 3000 | 3.14 | 1.62 | 2.66 | 3.03 | 8.85 | 2.83 | 4.34 | 4.02 | 8.71 |
| 3500 | 3.16 | 1.64 | 2.64 | 2.55 | 7.41 | 2.47 | 4.48 | 4.12 | 9.15 |
| 4500 | 3.16 | 1.62 | 2.66 | 2.54 | 7.14 | 2.47 | 4.49 | 4.12 | 9.16 |
| 5500 | 3.12 | 1.62 | 2.64 | 2.18 | 5.79 | 2.14 | 4.25 | 4.06 | 8.91 |
| 7000 | 3.12 | 1.56 | 2.66 | 1.96 | 5.31 | 1.92 | 3.99 | 3.96 | 8.65 |
| **Previous Results** | $r_{\text{Si-Si}}$ | 3.155 [15], 3.16 [16], 3.08 [17], 3.13 [18], 3.14 [19], 3.11 [23], 3.12 [21], 3.077 [22] | | | | | | | |
| | $r_{\text{Si-O}}$ | 1.595 [15], 1.63 [16], 1.62 [17], 1.61 [18], 1.61 [19], 1.60 [23], 1.62 [21], 1.608 [20] | | | | | | | |
| | $r_{\text{O-O}}$ | 2.59 [15], 2.62 [16], 2.66 [17], 2.65 [18], 2.60 [19], 2.61 [23], 2.65 [21], 2.626 [21] | | | | | | | |

**Table 3.** The number of structural units and link angles of bulk $SiO_2$ at different T values.

| T (K) | Structural Units Number | | | O-Si-O (Degree) | Results (Degree) Obtained Previously |
|---|---|---|---|---|---|
| | $SiO_4$ | $SiO_5$ | $SiO_6$ | | |
| 300 | 2975 | 121 | 7 | 105 | |
| 500 | 2071 | 153 | 7 | 105 | |
| 1500 | 2965 | 173 | 21 | 105 | 108.3 [15], 109 [17], |
| 2500 | 2959 | 232 | 0 | 105 | 107.3 [23] |
| 3000 | 2960 | 211 | 0 | 105 | by SM |
| 3500 | 2791 | 709 | 74 | 105 | and 109.47 [24], |
| 4500 | 2769 | 783 | 49 | 105 | 109.7 [25], 109.4 [26], |
| 5500 | 2653 | 959 | 119 | 100 | 109.5 [18] |
| 7000 | 2650 | 960 | 64 | 95 | by EM |

When (T) increases from T = 300 K to T = 500, 1500, 2500, 3000, 3500, 4500, 5500, 7000 K, the structure unit number $SiO_4$ decreases from 2975 atoms to 2650 atoms; whereas for $SiO_5$ it increases from 121 atoms to 960 atoms. Whereas for $SiO_6$ change in about from 0 atoms to 119 atoms (Table 3), the link angle of O-Si-O decreases from 105° to 95°. Our results are consistent with the results obtained previously. The links angle for O-Si-O [14,15,17,23] by SM, and by EM [21,24–26], are also in agreement with our calculations. The results show a significant influence of T on the link angle and the number of structural units $SiO_4$, $SiO_5$, $SiO_6$. The disappearance of the number of $SiO_6$ structural units at T = 2500, 3000 K shows that there is a phase transition of the material in this region. To answer this question, the l and total energy of the system ($E_{tot}$) at different T values (Table 4) have been also calculated.



**Table 4.** The l and $E_{tot}$ of bulk $SiO_2$ at different T values.

| T (K) | l (nm) | $E_{tot}$ (eV) |
|---|---|---|
| 300 | 3.440 | −53,230 |
| 500 | 3.442 | −53,072 |
| 1500 | 3.450 | −52,282 |
| 2500 | 3.451 | −51,477 |
| 3000 | 3.453 | −51,062 |
| 3500 | 3.454 | −50,501 |
| 4500 | 3.462 | −49,424 |
| 5500 | 3.473 | −48,258 |
| 7000 | 3.521 | −46,695 |

The obtained results show the case for the high T region T > 273 K. When T increases, $E_{tot}$ increases, and with T increasing from T = 300 K to T = 2500 K, the size (l) of the material increases greatly from l = 3.440 nm to 3.451 nm, and when T increases from T = 3000 K to T = 7000 K, l slows down from l = 3.453 nm to l = 3.521 nm (Table 4). The obtained results show that in the high T region, two regions appear, the crystalline state and the liquid state, in which the intersection region between the two liquid states and the crystalline state appears in the T range from T = 2500 K to T = 3000 K [19]. As a consequence, it can be generally said that the effect of T on the heterogeneous kinetics of the considered material is very large. The results obtained show that when T increases at P = 0 GPa, the links length (r) and angle of the links will not change significantly. With the first peak point of RDF g(r), the mean coordinate number tends to decrease, g(r) decreases strongly, and the link Si-O increases. This is the cause that leads to the insignificant change in the number of the structural unit of $SiO_4$, $SiO_5$, $SiO_6$, and the disappearance of structural units. The obtained results could serve as the basis for future experimental studies.

3.2.2. Low T Region

The results of structural characteristic quantities of the bulk $SiO_2$ at low T values are presented in Table 5.

**Table 5.** The structural characteristic quantity of bulk $SiO_2$ at low T values.

| T(K) | Links Lengths $r_{ij}$ (Å) | | | First Peak Positions $g(r_{ij})$ | | | CN | | | l (nm) | Etot (eV) |
|---|---|---|---|---|---|---|---|---|---|---|---|
| | Si-Si | Si-O | O-O | Si-Si | Si-O | O-O | Si-Si | Si-O | O-O | | |
| 300 | 3.16 | 1.62 | 2.64 | 4.47 | 24.76 | 4.78 | 4.11 | 4.01 | 8.51 | 3.439 | −53,230 |
| 194.5 | 3.18 | 1.64 | 2.64 | 4.60 | 28.36 | 5.00 | 4.13 | 4.01 | 8.41 | 3.439 | −53,312 |
| 90 | 3.2 | 1.64 | 2.64 | 4.63 | 35.28 | 5.22 | 4.12 | 4.01 | 10.2 | 3.439 | −53,394 |
| 83.8085 | 3.2 | 1.64 | 2.64 | 4.64 | 35.81 | 5.22 | 4.12 | 4.01 | 6.73 | 3.439 | −53,399 |
| 70 | 3.2 | 1.64 | 2.64 | 4.61 | 37.26 | 5.27 | 4.14 | 4.01 | 6.73 | 3.439 | −53,410 |
| 4.22 | 3.22 | 1.64 | 2.62 | 4.67 | 42.51 | 5.57 | 4.13 | 4.00 | 7.78 | 3.439 | −53,460 |
| **The number of structural units and bond angle at low T values** | | | | | | | | | | | |
| T(K) | 300 | 194.5 | 90 | 83.8085 | 70 | 4.22 | | | | | |
| $SiO_4$ | 2975 | 2974 | 2970 | 2969 | 2970 | 2974 | | | | | |
| $SiO_5$ | 121 | 128 | 144 | 150 | 144 | 133 | | | | | |
| $SiO_6$ | 7 | 7 | 7 | 7 | 7 | 7 | | | | | |
| O-Si-O (degrees) | 105 | 105 | 105 | 105 | 105 | 105 | | | | | |

The obtained results show that with bulk $SiO_2$ at (T), T = 300 K has structural characteristic quantities (Figure 1). The results obtained in the low T region T < 273 K (negative T values), P = 0 GPa. When the T decreases from T = 300 K to T = 194.5, 90, 83.8085, 70, 4.22 K the lengths links of Si-Si, Si-O, O-O have changed as Si-Si increases (from $r_{Si-Si}$ = 3.16 Å to $r_{Si-Si}$ = 3.22 Å), Si-O increases (from $r_{Si-O}$ = 1.62 Å to $r_{Si-O}$ = 1.64 Å) and O-O decreases (from $r_{O-O}$ = 2.64 Å to $r_{O-O}$ = 2.62 Å); CN changes very little, l is constant value l = 3.439 nm; energy increased (from $E_{tot}$ = −53,230 eV to $E_{tot}$ = −53,460 eV); the number of $SiO_4$, $SiO_5$, $SiO_6$ structural units has a constant value, and the O-Si-O bond angle has a constant value of 105° (Table 5) which shows that in this region there is almost no structural change, but there is an increase in energy of the $E_{tot}$ system. With this, the researchers can use this material in future energy storage devices.

### 3.2.3. Effects of P

As has been emphasized in the Introduction, other scientists only considered the effect of P at T = 300 K, and there are no research results in the high T area. Therefore, the study on the structural characteristics and phase transition of $SiO_2$ in the high T and high P regions has also been carried out and presented in this text.

### At T = 70 K

The structural characteristic quantities of bulk $SiO_2$ at T = 70 K with different *p* values are shown in Table 6.

**Table 6.** The structural characteristic quantity of bulk $SiO_2$ at low T values.

| P (GPa) | Links Lengths $r_{ij}$ (Å) | | | First Peak Positions $g(r_{ij})$ | | | CN | | | l (nm) | $E_{tot}$ (eV) |
|---|---|---|---|---|---|---|---|---|---|---|---|
| | Si-Si | Si-O | O-O | Si-Si | Si-O | O-O | Si-Si | Si-O | O-O | | |
| 0 | 3.20 | 1.64 | 2.64 | 4.61 | 37.26 | 5.27 | 4.14 | 4.01 | 6.73 | 3.439 | −53,410 |
| 5 | 3.12 | 1.62 | 2.62 | 3.79 | 35.11 | 4.58 | 4.41 | 4.04 | 7.57 | 3.317 | −53,347 |
| 10 | 3.06 | 1.62 | 2.62 | 3.64 | 27.69 | 4.29 | 4.82 | 4.09 | 8.26 | 3.232 | −53,219 |
| 15 | 3.06 | 1.62 | 2.56 | 3.26 | 16.54 | 3.62 | 5.88 | 4.34 | 9.33 | 3.111 | −52,990 |
| 20 | - | - | - | - | - | - | - | - | - | - | - |

| The number of structural units and bond angle at low T | | | | | |
|---|---|---|---|---|---|
| P (GPa) | 0 | 5 | 10 | 15 | 20 |
| $SiO_4$ | 2970 | 2903 | 2697 | 1990 | - |
| $SiO_5$ | 144 | 389 | 894 | 1725 | - |
| $SiO_6$ | 7 | 35 | 120 | 757 | - |
| O-Si-O (degree) | 105 | 105 | 105 | 105 | - |

The results obtained at T = 70 K, P = 0 GPa show that when the P increases from P = 0 GPa to P = 0, 5, 10, 15, 20 GPa the lengths links of Si-Si, Si-O, O-O have changed as Si-Si decreases (from $r_{Si-Si}$ = 3.20 Å to $r_{Si-Si}$ = 3.06 Å), Si-O decreases (from $r_{Si-O}$ = 1.64 Å to $r_{Si-O}$ = 1.62 Å) and O-O decreases (from $r_{O-O}$ = 2.64 Å to $r_{O-O}$ = 2.56 Å); CN increases from $CN_{Si-Si}$ = 4.14 to 5.88, $CN_{Si-O}$ = 4.01 to 4.34, $CN_{O-O}$ = 6.73 to 9.33, l is decreased from l = 3.439 nm to l = 3.111 nm; energy increased (from $E_{tot}$ = −53,410 eV to $E_{tot}$ = −52,990 eV); the number of structural units of $SiO_4$ decreases from 2970 atoms to 1990 atoms, $SiO_5$ increased from 144 atoms to 1725 atoms, $SiO_6$ increased from 7 atoms to 757 atoms, and the O-Si-O bond angle has a constant value of 105°, in which the number of $SiO_4$, $SiO_5$, and $SiO_6$ structural units disappear at P = 20 GPa, showing that bulk $SiO_2$ materials at low T only exist in the region of P < 20 GPa (Table 6), which shows that in this region there is almost no structural change, but there is an increase in energy of the $E_{tot}$ system. Based on these results, the researchers can use this material in future energy storage devices.

At T = 300 K

The structural characteristic quantities of bulk SiO$_2$ at T = 300 K with different *p* values are shown in Figure 2.

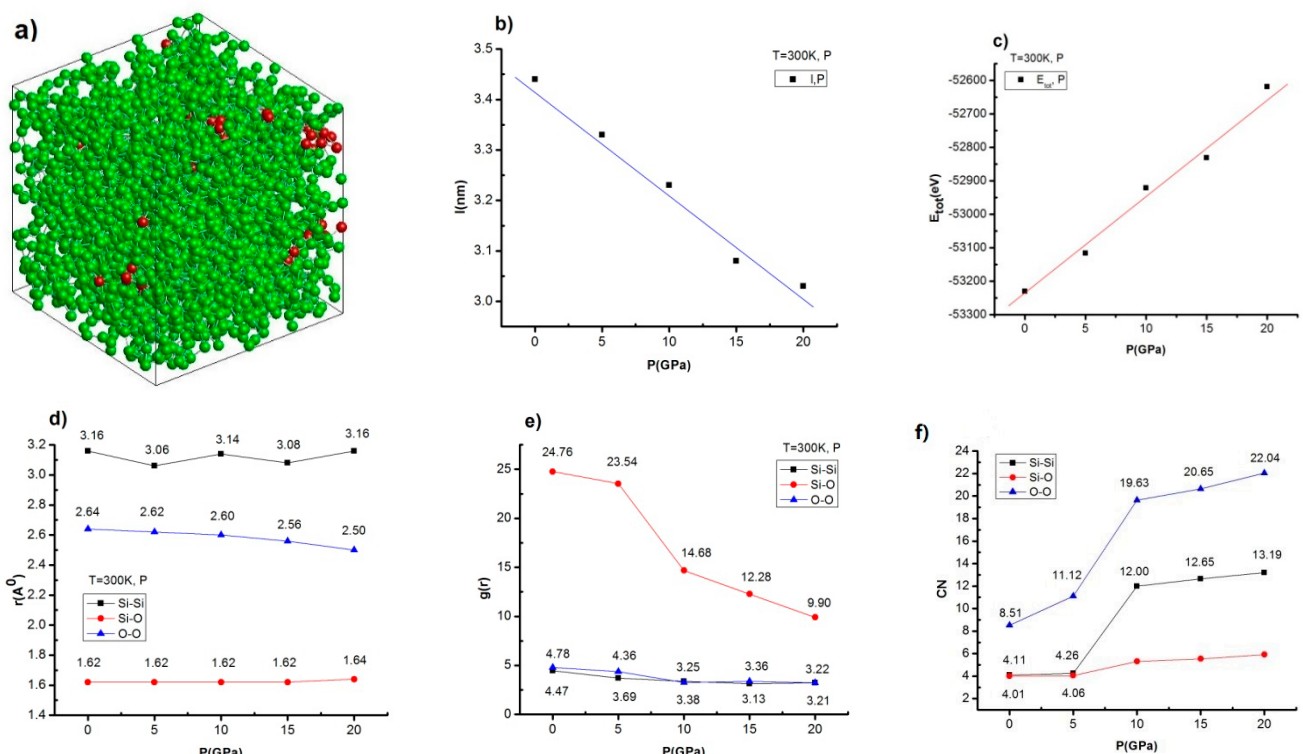

**Figure 2.** Shape (**a**), l (**b**), the Etot of the system (**c**), r (**d**), g(r) (**e**), coordinate number (**f**) of bulk SiO2 at T = 300 K with different values of *p*.

The obtained results demonstrate the fact that bulk SiO$_2$ at T = 300 K, P = 0 GPa has the form of a cube (Figure 2a) with the nanoscale, l = 3.44 nm, and the E$_{tot}$ is equal to −53,230 eV. The form of RDF gives the lengths of the links (r) of Si-Si, Si-O, O-O equal to r$_{Si-Si}$ = 3.16 Å, r$_{Si-O}$ = 1.62 Å, r$_{O-O}$ = 2.64 Å. These correspond to the heights of RDF g$_{Si-Si}$ = 4.47, g$_{Si-O}$ = 24.76, g$_{O-O}$ = 4.78. The average CNs were calculated by the formula (4) and have the following values: CN$_{Si-Si}$ = 4.11, CN$_{Si-O}$ = 4.01, CN$_{O-O}$ = 8.51, respectively. The number of structural units SiO$_4$, SiO$_5$, SiO$_6$ are correspondingly 1978, 63, and 1, whereas O-Si-O links angle is 105° and Si-O-Si is 140°. When P increases from P = 0 GPa to 5, 10, 15, 20 GPa, the size (l) decreases from l = 3.44 nm to l = 3.33, 3.23, 3.08, 3.03 nm (Figure 2b), which correspond to E$_{tot}$ increasing from E$_{tot}$ = −53,230 eV to E$_{tot}$ = −53,116, −52,921, −52,831, −52,619 eV (Figure 2c), respectively. RDF had the position r slightly changed (from 3.08 Å to 3.16 Å for Si-Si), (1.62 Å to 1.64 Å for Si-O), and (2.50 Å to 2.64 Å for O-O) (Figure 2d), which corresponds to a decrease of g(r) (from 4.47 to 3.21 with Si-Si), (24.76 to 9.90 with Si-O), (4.78 to 3.22 with O-O) (Figure 2e).

The average CNs (4) increased from 4.11 to 13.19 for Si-Si, 4.01 to 5.91 with Si-O, 8.51 to 22.04 with O-O (Figure 2f), while the number of SiO$_4$ structural units decreased from 2975 atoms to 2906, 1928, 1536, 983 atoms. This number of SiO$_5$ increases from 121 atoms to 345, 1855, 2070, and 2163 atoms, whereas for SiO$_6$ it increases from seven atoms to 32, 640, 932, and 1450 atoms. The O-Si-O links angle remains constant at 105°.

The results obtained show that when P increases at T = 300 K, the length (r), g(r), CN, and angle of the links will strongly change. With r$_{O-O}$, g$_{Si-O}$ strongly decreases and CN$_{O-O}$, CN$_{Si-Si}$ strongly increases, and this leads to a sudden decrease in the number of SiO$_4$ structural units and the sudden increase of SiO$_5$, SiO$_6$.

At T = 1273 K

The structural characteristic quantities of bulk SiO$_2$ at T = 1273 K with different *p* values are shown in Figure 3.

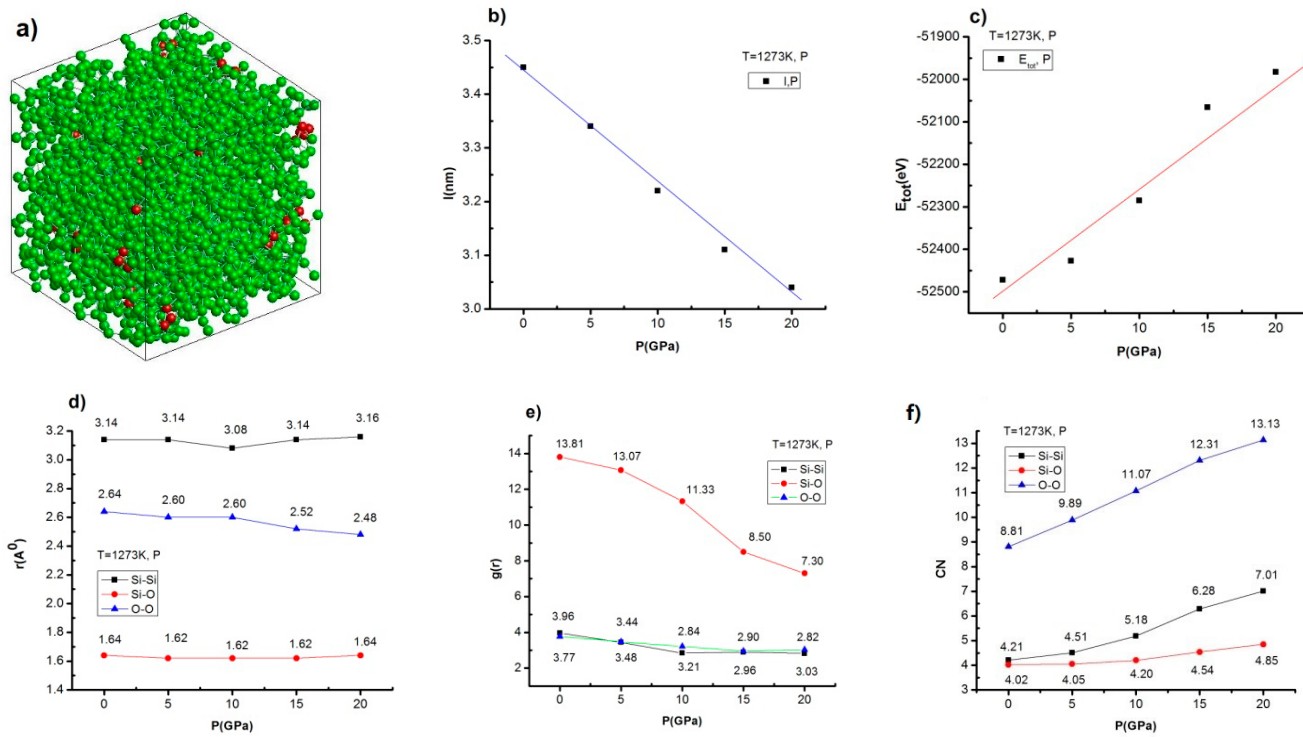

**Figure 3.** Shape (**a**), l (**b**), Etot (**c**), r (**d**), g(r) (**e**), coordinate number (**f**) of bulk SiO2 at T = 1273 K with differ-ent values of *p*.

The results are presented similarly to the previous case of T = 300 K. Namely, bulk SiO$_2$ at T = 1273 K, P = 0 GPa has the form of a cube (Figure 3a) with the nanoscale, l = 3.45 nm, the E$_{tot}$ is −52,472 eV; RDF with links length (r) of Si-Si, Si-O, O-O r$_{Si-Si}$ = 3.14 Å, r$_{Si-O}$ = 1.64 Å, r$_{O-O}$ = 2.64 Å corresponding to the height of RDF is g$_{Si-Si}$ = 3.96, g$_{Si-O}$ = 13.81, g$_{O-O}$ = 3.77. The average CNs are CN$_{Si-Si}$ = 4.21, CN$_{Si-O}$ = 4.02, CN$_{O-O}$ = 8.81. When P increases from P = 0 GPa to 5, 10, 15, 20 GPa, (l) decreases from l = 3.45 nm to l = 3.34, 3.22, 3.11, 3.04 nm (Figure 3b) which correspond to the increase of E$_{tot}$ from E$_{tot}$ = −52,472 eV to E$_{tot}$ = −52,427, −52,285, −52,066, −51,983 eV (Figure 3c). RDF has very large change of r from 3.08 Å to 3.16 Å for Si-Si, 1.62 Å to 1.64 Å for Si-O; 2.48 Å to 2.64 Å with O-O (Figure 3d). These correspond to a decrease of g(r) from 3.96 to 2.82 for Si-Si, from 13.81 to 7.30 with Si-O, and from 3.77 to 3.03 with O-O (Figure 3e). The average CN increases from 4.21 to 7.01 with Si-Si, from 4.02 to 4.85 with Si-O, and from 8.81 to 13.13 with O-O (Figure 3f), which corresponds to the decrease in structural units number for SiO$_4$ from 2970 atoms to 2900, 2571, 1685, 1046 atoms. For SiO$_5$ this number increases from 147 atoms to 398, 1199, 1961, 2042 atoms, and for SiO$_6$ it increases from 0 atoms to 36, 205, 881, 1474 atoms. The O-Si-O links angle remains constant at 105°. The results obtained show that when P increases at T = 1273 K, the length (r), g(r), CN, and angle of the links will change. Also, r$_{O-O}$, g$_{Si-O}$ decrease, and CN$_{O-O}$, CN$_{Si-Si}$ increase. This leads to a sudden slow-down decrease in the number of SiO$_4$ structural units and the slow-down increase of SiO$_5$, and SiO$_6$.

At T = 2974 K

The structural characteristic quantities of bulk SiO$_2$ at T = 2974 K with different values of *p* are demonstrated in Figure 4.

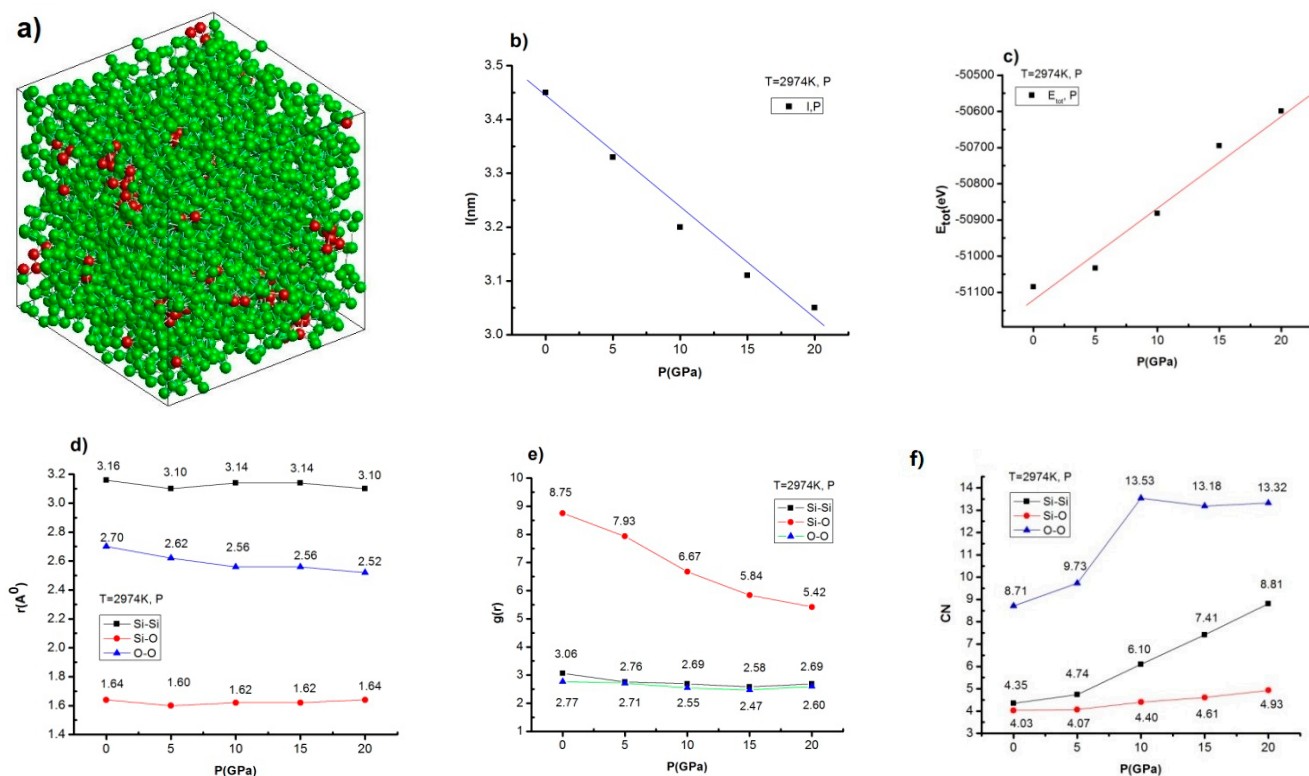

**Figure 4.** Shape (**a**), l (**b**), Etot (**c**), r (**d**), g(r) (**e**), coordinate number (**f**) of bulk SiO2 at T = 2974 K with differ-ent values of *p*.

It follows from these results that bulk $SiO_2$ at T = 2974 K, P = 0 GPa has the form of a cube (Figure 4a) with the size l = 3.45 nm, $E_{tot}$ = −51,085 eV. The lengths of the links of Si-Si, Si-O, O-O are $r_{Si-Si}$ = 3.16 Å, $r_{Si-O}$ = 1.64 Å, $r_{O-O}$ = 2.70 Å what corresponds to the height in the peak position of RDF $g_{Si-Si}$ = 3.06, $g_{Si-O}$ = 8.75, $g_{O-O}$ = 2.77. The average CNs are $CN_{Si-Si}$ = 4.35, $CN_{Si-O}$ = 4.03, $CN_{O-O}$ = 8.71. When P increases from P = 0 GPa to 5, 10, 15, 20 GPa, the size l decreases from l = 3.45 nm to l = 3.33, 3.20, 3.11, 3.05 nm (Figure 4b). This corresponds to increase of $E_{tot}$ from $E_{tot}$ = −51,085 eV to $E_{tot}$ = −51,033, −50,882, −50,695, −50,599 eV (Figure 4c). RDF has larger, r changes between 3.10 Å to 3.16 Å for Si-Si, 1.60 Å to 1.64 Å with Si-O, for 2.52 Å to 2.70 Å with O-O (Figure 4d). This corresponds to a decrease of g(r) from 2.58 to 3.06 with Si-Si, 5.42 to 8.75 with Si-O, 2.47 to 2.77 with O-O (Figure 4e). The CN increases from 4.35 to 8.81 with Si-Si, from 4.03 to 4.93 with Si-O, and from 8.71 to 13.32 with O-O (Figure 4f) which corresponds to the decrease in the number of structural units for $SiO_4$ from 2938 atoms to 2984, 2211, 1434, 961 atoms. This number for $SiO_5$ increases from 295 atoms to 713, 1660, 2079, 2138 atoms, whereas for $SiO_6$ it increases from 14 atoms to 111, 413, 1019, 1501 atoms, and $SiO_7$ it increases from 0 atoms to 72 atoms.

It can be seen that the $SiO_7$ structure unit number appears at P = 20 GPa; this result is completely consistent with the result of bulk $Fe_2O_3$ when P is increased at the high T region. The O-Si-O links angle remains constant at 105°. The results obtained show that when P increases at T = 2974 K, the length (r), g(r), CN, and angle of the links will change. Also, $r_{O-O}$, $g_{Si-O}$ decrease and $CN_{O-O}$, $CN_{Si-Si}$ increase; this leads to a sudden slow down decrease in the number of $SiO_4$ structural units and a very big increase of this number for $SiO_5$, $SiO_6$.

At T = 3500 K

The structural characteristic quantities of bulk $SiO_2$ at T = 3500 K with different values of *p* are shown in Figure 5.

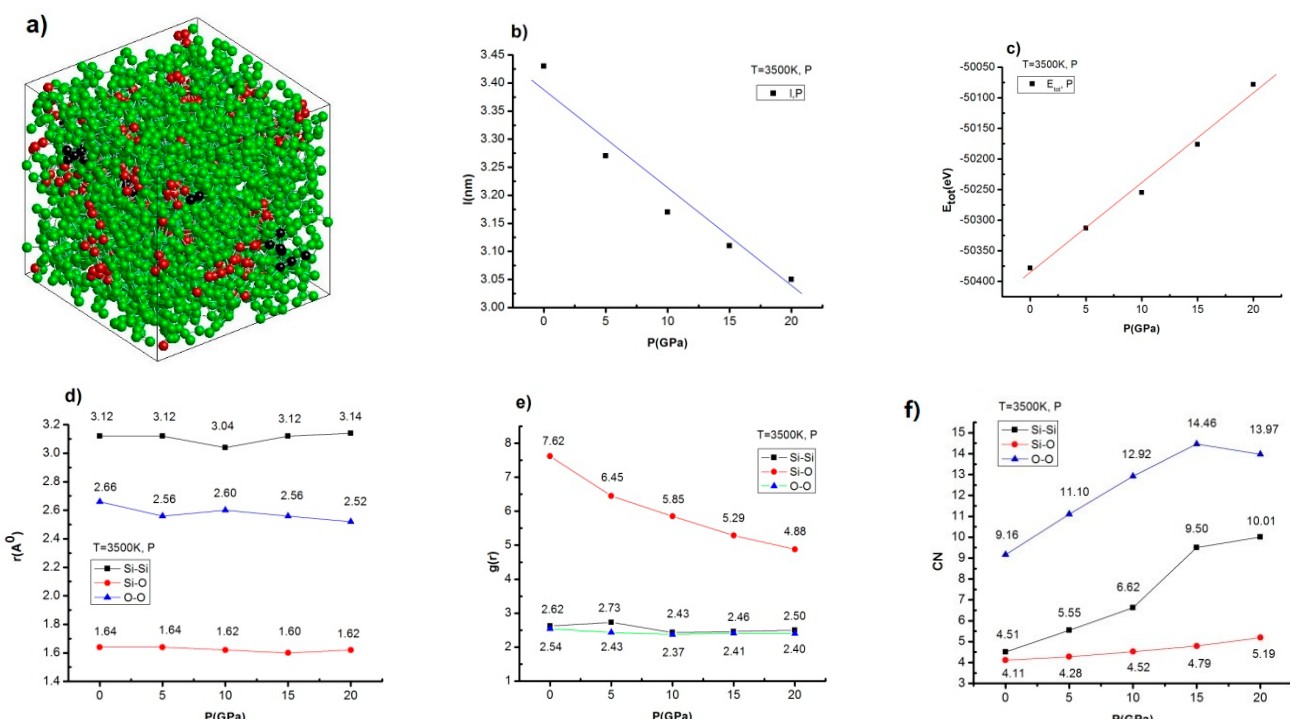

**Figure 5.** Shape (**a**), l (**b**), Etot (**c**), r (**d**), g(r) (**e**), coordinate number (**f**) of bulk SiO2 at T = 3500 K with differ-ent values of *p*.

The calculated results show, that bulk $SiO_2$ at T = 3500 K, P = 0 GPa has also the form of a cube (Figure 5a) with the nanoscale, l = 3.43 nm. The $E_{tot}$ is equal to $-50,378$. The RDF gives the links lengths (r) of Si-Si, Si-O, O-O which are $r_{Si-Si}$ = 3.12 Å, $r_{Si-O}$ = 1.64 Å, $r_{O-O}$ = 2.66 Å. This corresponds to the height of the first peak position of RDF $g_{Si-Si}$ = 2.62, $g_{Si-O}$ = 7.62, $g_{O-O}$ = 2.54. The average CNs are $CN_{Si-Si}$ = 4.51, $CN_{Si-O}$ = 4.11, $CN_{O-O}$ = 9.16. When P increases from P = 0 GPa to 5, 10, 15, 20 GPa, the size (l) decreases from l = 3.43 nm to l = 3.27, 3.17, 3.11, 3.05 nm (Figure 5b). This corresponds to increase of $E_{tot}$ from $E_{tot}$ = $-50,378$ eV to $E_{tot}$ = $-50,313$, $-50,255$, $-50,176$, $-50,078$ eV (Figure 5c), relatively. RDF has a very large r changes from 3.04 Å to 3.14 Å for Si-Si, from 1.60 Å to 1.64 Å for Si-O, and from 2.52 Å to 2.66 Å with O-O (Figure 5d). These changes correspond to a decrease of g(r) from 2.73 to 2.43 with Si, from 7.62 to 4.88 with Si-O and from 2.54 to 2.37 with O-O (Figure 5e). The average CNs increase from 4.51 to 10.01 with Si-Si, from 4.11 to 5.19 with Si-O, and from 9.16 to 14.46 with O-O (Figure 5f) which corresponds to the decrease of the number of structural units for $SiO_4$ from 2820 atoms to 2373, 1788, 1391, 887 atoms. This number for $SiO_5$ increases from 632 atoms to 1520, 1935, 2100, 2150 atoms, whereas for $SiO_6$ it increases from 62 atoms to 222, 742, 1014, 1448 atoms, and $SiO_7$ it increases from 0 atoms to 173 atoms. In which, the appearance of structural units number of $SiO_7$ at P > 15 GPa with T = 3500 K. The O-Si-O links angle changes from 105° to 100°. The O-Si-O links angle changes from 105° to 100°. It can be concluded that when P increases, l decreases, $E_{tot}$ increases, r changes, and g(r) decreases with Si-O, but it changes insignificantly for Si-Si, O-O. The $CN_{Si-O}$ changes in such a way that the number of $SiO_4$ structural units decreases, whereas for $SiO_5$, $SiO_6$ increases with constant links angle O-Si-O equal to 105°. The results obtained show, that when P increases at T = 3500 K, the length (r), g(r), CN, and angle of the links will change. Also, $r_{O-O}$, $g_{Si-O}$ decrease, and $CN_{O-O}$, and $CN_{Si-Si}$ increase. This leads to a sudden slow-down decrease in the number of $SiO_4$ structural units and the slow-down increase of $SiO_5$, and $SiO_6$. When P = 0 GPa and T increase, the numbers of structural units of $SiO_4$, $SiO_5$, and $SiO_6$ have no significant change. When P increases at T = 300, 1273, 2974, 3500 K, the number of structural units $SiO_4$ decreases, while for $SiO_5$, $SiO_6$ increases and there is the largest change at T = 2974 K. This fact proves that at $T_m$ = 2974 K, the

largest change in the number of structural units exists and this will be the basis for future experimental studies.

## 4. Conclusions

In this study, using the SM of MD, the effects of T and P on the heterogeneous kinetics of the bulk $SiO_2$ (with 3000 atoms at 300 K, 500 K, 1000 K, 1500 K, 2000 K, 2500 K, 3000 K, 3500 K, 4500 K, 5500 K, and 7000 K; at 0 GPa, 5 GPa, 10 GPa, 15 GPa, and 20 GPa with T = 300 K, 1273 K, 2974 K, 3500 K) have been considered. These considerations lead to a conclusion that with bulk $SiO_2$ (3000 atoms), the choice BKS potentials gives the results consistent with previous both experimental and simulation results. The increase in the T leads to the initial increase in the l. The $E_{tot}$ increases gradually as the T increases at P = 0 GPa. It follows from the obtained results that for the T range from T = 300 K to T = 2974 K, bulk $SiO_2$ exists in an amorphous state, whereas for T > 2974 K bulk $SiO_2$ exists in a liquid state, so the $T_m$ of bulk $SiO_2$ has been determined as 2974 K. When T increases from P = 0 GPa to P = 5, 10, 15, 20 GPa with T = 300, 1273, 2974, 3500 K calculated the structural units number, which for $SiO_4$ it decreases, while for $SiO_5$, $SiO_6$ it increases, while the number of $SiO_7$ structural units appears with P > 15 GPa at T = 3500 K, P > 20 GPa at 2974 K. The l of bulk $SiO_2$ decreases and $E_{tot}$ increases, g(r) of Si-O decreases, CN of Si-Si, O-O increases strongly with higher T. It follows from these results that for low T, the CN changes very strongly, while T is large, T > 2974 K, CN changes insignificantly. Our results show generally that there is a significant influence of T and P on the structure and phase transition of bulk $SiO_2$. These are new results which could serve as a basis for future experimental studies.

**Author Contributions:** D.N.T.: Conceptualization, Methodology, Investigation, Validation, Writing-original draft-review & editing, Formal analysis. V.C.L.: Writing- original draft; Formal analysis. Ş.Ţ.: Writing-original draft & editing. U.S.: Writing-original draft; Formal analysis. P.N.D.: Writing-original draft; Formal analysis. K.P.H.: Writing-original draft; Formal analysis. All authors have read and agreed to the published version of the manuscript.

**Funding:** There is no source of funding to support publishing fees.

**Institutional Review Board Statement:** Not applicable.

**Informed Consent Statement:** Not applicable.

**Data Availability Statement:** The data that support the findings of this study are available from the corresponding author upon reasonable request.

**Acknowledgments:** This research was funded by the Thai Nguyen University under grant number ĐH2022-TN04-02.

**Conflicts of Interest:** The authors declare that they have no conflict of interest.

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
