# Peer review of "A New Study on the Structure, and Phase Transition Temperature of Bulk Silicate Materials by Simulation Method of Molecular Dynamics"

_jcs, doi:10.3390/jcs6080234_

Round 1

Reviewer 1 Report

Dear Authors, the article starts with too many self citation (the first 27 references, except one, are self citation. Therefore, it may be seen like a review paper of book chapter. In my opinion, in the introduction, we must have seen the comparison with the international literature in this subject.

Author Response

Review 1

Dear Authors, the article starts with too many self citation (the first 27 references, except one, are self citation. Therefore, it may be seen like a review paper of book chapter. In my opinion, in the introduction, we must have seen the comparison with the international literature in this subject.

Answer

Thank you very much, review for taking the time to read our manuscript and for your very helpful comments that gave us the opportunity to edit and improve the content of the article. In your opinion, self-quoting our 27 references is unnecessary and inconsistent with the content of the article. In the first part of the introduction, we said very clearly that the study of materials from metals, alloys, polymer materials is related to the structure and phase transition of materials. In response to the review's request, we have removed the documents that the review considers unrelated to the topic, edited the entire English style, grammar and hopefully with the content. This revision will meet the review's and journal's requirements.

Reviewer 2 Report

Please find my comments below

(1) Line 54 is incomplete

(2) Sadly, the whole introduction part is written with poor background, structuring and paragraphing.

(3) Plagiarism detected attached.

(4) The work has been extensively worked upon even by the authors, so i do not see any need making it less significant. Even though, the work done should not be compare with the authors' previous work but open literature.

(5) Please change the title to: A study on the structure and phase transition temperature of bulk silicate materials using molecular dynamics.

Author Response

Review 2

Answer

Thank you very much for taking the time to read our manuscript and for your very helpful comments that gave us the opportunity to edit and improve the content of the article. Here is our feedback

 (1) Line 54 is incomplete

Answer

+We have finished editing

(2) Sadly, the whole introduction part is written with poor background, structuring and paragraphing.

Answer

+ The author has revised the style, writing style and hopes to meet your requirements, all edits have been highlighted in red for easy tracking.

(3) Plagiarism detected attached.

Answer

+ The author is very surprised, I don't understand where you said we plagiarized, this is the content of our new article, we have never posted it online anywhere. Maybe the review is suspicious of links on the internet.

I quote for the review's review:

-https://www.researchgate.net/publication/319464912_The_influence_of_temperature_on_the_microstructure_and_the_phase_transition_process_of_the_SiO2_bulk_model

This is the route that one of our co-authors previously put up to promote the article content before submitting it, we fixed it by removing the title of this article and asked the editor of this page deletes it for me, we guarantee that this is a completely new article content, has not been posted anywhere online and is fully responsible for the content of the article for the review to review.

(4) The work has been extensively worked upon even by the authors, so i do not see any need making it less significant. Even though, the work done should not be compare with the authors' previous work but open literature.

Answer

+The author has removed comparisons with previously published results, hoping to satisfy your request

(5) Please change the title to: A study on the structure and phase transition temperature of bulk silicate materials using molecular dynamics.

Answer

+ The author agrees to change the title of the article to "A study on the structure and phase transition temperature of bulk silicate materials using molecular dynamics "

All edits by the author will hopefully meet the review's request.

Round 2

Reviewer 1 Report

The authors improved the introduction, as suggested.

Reviewer 2 Report

Dear Authors,

Thank you for significantly improving on the manuscript. I feel satisfied with its content. Thank you and wishing you the best.